# Developing a video expert panel as a reference standard to evaluate respiratory rate counting in paediatric pneumonia diagnosis: protocol for a cross-sectional study

Ahad Mahmud Khan ![ORCID],[1,2] Salahuddin Ahmed ![ORCID],[1,2] Nabidul Haque Chowdhury,[2] Md Shafiqul Islam,[2] Eric D McCollum ![ORCID],[3,4] Carina King,[5] Ting Shi ![ORCID],[1] Kamrun Nahar,[6] Robynne Simpson,[7] Ayaz Ahmed,[7] Md Mozibur Rahman,[8] Abdullah H Baqui,[4] Steve Cunningham,[9] Harry Campbell,[1] RESPIRE Collaboration

For numbered affiliations see end of article.

**Correspondence to**
Dr Ahad Mahmud Khan;
ahad_mahmud@hotmail.com

## ABSTRACT

**Introduction**  Manual counting of respiratory rate (RR) in children is challenging for health workers and can result in misdiagnosis of pneumonia. Some novel RR counting devices automate the counting of RR and classification of fast breathing. The absence of an appropriate reference standard to evaluate the performance of these devices is a challenge. If good quality videos could be captured, with RR interpretation from these videos systematically conducted by an expert panel, it could act as a reference standard. This study is designed to develop a video expert panel (VEP) as a reference standard to evaluate RR counting for identifying pneumonia in children.

**Methods and analysis**  Using a cross-sectional design, we will enrol children aged 0–59 months presenting with suspected pneumonia at different levels of health facilities in Dhaka and Sylhet, Bangladesh. We will videorecord a physician/health worker counting RR manually and also using an automated RR counter (Children's Automated Respiration Monitor) from each child. We will establish a standard operating procedure for capturing quality videos, make a set of reference videos, and train and standardise the VEP members using the reference videos. After that, we will assess the performance of the VEP as a reference standard to evaluate RR counting. We will calculate the mean difference and proportions of agreement within ±2 breaths per minute and create Bland-Altman plots with limits of agreement between VEP members.

**Ethics and dissemination**  The study protocol was approved by the National Research Ethics Committee of Bangladesh Medical Research Council, Bangladesh (registration number: 39315022021) and Edinburgh Medical School Research Ethics Committee (EMREC), Edinburgh, UK (REC Reference: 21-EMREC-040). Dissemination of the study findings will be through conference presentations and publications in peer-reviewed scientific journals.

### STRENGTHS AND LIMITATIONS OF THIS STUDY

⇒ Expert paediatricians will provide feedback to develop a standard operating procedure for videography of child chest movements.
⇒ The video expert panel will be trained and standardised using the expert paediatrician-interpreted reference videos.
⇒ Video expert panel members will be masked to respiratory rate counted by each other and to respiratory rate manual counts, and with the automated counter.
⇒ Children with varying severity of illness will be enrolled from different levels of health facilities in Bangladesh.
⇒ Despite the availability of multiple respiratory rate counters (eg, Children's Automated Respiration Monitor, ChARM, Rad-G, uPM60), only the ChARM device will be used, which is a limitation in this study.

worldwide.[1] Approximately 68 million pneumonia episodes and 0.65 million deaths occur annually in under-5 children due to pneumonia.[2] The highest number of deaths due to pneumonia among children below 5 years were in sub-Saharan Africa, South Asia and Southeast Asia. There were an estimated 2.7 million episodes of pneumonia in Bangladesh and 21 275 deaths due to childhood pneumonia in 2015.[3]

According to the WHO guidelines, pneumonia diagnosis in children for frontline health workers is primarily based on increased respiratory rate (RR) and/or chest indrawing.[4 5] Fast breathing is the most common sign of pneumonia. It is most commonly identified by counting the RR manually.[5] However, manual counting can

## INTRODUCTION
Pneumonia is one of the leading causes of mortality in children aged below 5 years

be challenging for the health workers,[6] often leading to incorrect diagnosis and, consequently, inappropriate treatment.[7–9]

Better performing RR diagnostics to support frontline health workers might improve the diagnosis of pneumonia. Three automated RR counting devices are considered suitable for use by health workers, that is, Children's Automated Respiration Monitor (ChARM),[10 11] Rad-G[12 13] and uPM60.[14] The Philips ChARM converts chest movements detected by accelerometers into a precise breathing count using specially designed algorithms. The device is placed around the child's belly, and it automatically counts the number of respirations for a particular period, calculates RR and classifies fast breathing according to WHO guidelines.[10 11] The usability and acceptability of this device have been tested in a study in Ethiopia[15] and Nepal.[16]

A reference standard is essential to evaluate the performance of RR counters in field settings. However, the absence of an appropriate reference standard is a challenge. Different reference standards have been used in various studies.[17 18] Most of the existing studies have used manual RR count by an expert person (eg, physician, nurse) as the reference standard.[19] Few studies have been found using automated monitors.[20 21] The possible biases using a human expert's count as the reference standard include difficulty in measuring the RR over the same simultaneous period and inconsistencies in human expert RR counting.[6] The automated monitors do not measure chest movements directly, but other variables such as carbon dioxide, pulse oximeter signal or photoplethysmogram, sound etc, are used to extract RR.[19 22–24]

Videography of a child's chest movements and interpretation by the experts could be used as a reference standard.[17 18 25] If quality videos could be recorded and the interpretation of RR from these videos could be systematically conducted by a video expert panel (VEP), it could be an ideal and non-biased reference standard. This study aims to develop a VEP as a reference standard for evaluating RR counting manually and using ChARM in paediatric pneumonia diagnosis. We will establish a standard operating procedure (SOP) for capturing quality videos, make a set of reference videos, and train and standardise the VEP members using the reference videos. The SOP, recorded videos and evaluation methods could be used for evaluating new RR counting devices.

## Study objectives
### Primary objectives
1. To develop an SOP for video recording a child's chest movements for RR interpretation.
2. To create a set of reference videos for the training and standardisation of the VEP members.
3. To train and standardise the VEP members to interpret RR using the reference videos.
4. To assess the performance of the VEP to evaluate RR counting manually and with ChARM based on videos captured in real-world settings.

### Secondary objectives
1. To assess the accuracy of the ChARM device in counting RR compared with the VEP as the reference.
2. To determine the duration of counting RR using the ChARM device.
3. To explore the potential influence of the ChARM device on RR count compared with standard observation techniques.
4. To assess the agreement of manual RR count by healthcare staff with RR counted by VEP.

## METHODS AND ANALYSIS
### Study design and settings
This will be a cross-sectional study. First, we will capture videos of child chest movements, develop an SOP of videography, make a set of reference videos and use these reference videos to train and standardise the VEP. We will record these videos from the inpatient department (IPD) of the Institute of Child and Mother Health (ICMH), Dhaka, Bangladesh where children with different severity of illness are admitted. After that, we will use the VEP as a reference standard to evaluate RR counting. The videos will be recorded from different levels of health facilities in Bangladesh. Three community clinics (CCs) and a subdistrict hospital (Zakiganj Upazila Health Complex) in Sylhet and ICMH in Dhaka will be selected. CCs are the lower-level health facility in Bangladesh staffed by community healthcare providers (CHCP).[26] Patients from the CCs are referred to subdistrict hospitals. Patients often go to subdistrict hospitals directly.[27] The study started on 6 December 2021 and is planned to be completed by December 2022.

### Study population
#### Inclusion criteria
1. Age <2 months presenting with any illness.
2. Age 2–59 months presenting with cough and/or difficulty breathing.

#### Exclusion criteria
1. Presence of any danger sign (unable to drink or feed, vomit everything, convulsions, lethargic or unconscious).
2. The parents are unwilling to provide consent.

### Study procedures
#### Enrolment and consenting
In the hospital, a physician will screen all children who will visit the outpatient clinic (subdistrict hospital) or are already admitted to IPD (ICMH) and enrol eligible children. In the CC, the CHCP will screen every child who will visit his/her CC. If a child becomes eligible, the CHCP will enrol the child and contact research staff who will visit the CC immediately and will complete data collection. Informed written consent will be obtained from the child's mother, father or other available caregiver before data collection.

## Developing SOP of videography

Research staff will record videos of child chest movements from 30 children. The Canon EOS M50 camera[28] will be used for recording videos. If a good source of natural light is not available, we will use a light-emitting diode lamp for lighting. The child's clothes will be removed from the lower end of the neck through the umbilicus so that the chest and belly are exposed. The child's face or any other identifiable part will not be recorded. The videography procedure (eg, camera positions, angles and levels of light, use of tripod) will be varied between children depending on the child and environmental factors (eg, child condition, position, adequate exposure, place of assessment). Videorecording will be done both when counting RR manually twice and using the ChARM device twice.

Once the child is calm, the research staff will start videorecording, and the physician will tap the microphone with his/her finger and begin counting RR. The physician will count RR for 1 min and tap the microphone again when counting is finished, and the research staff will stop the recording. After that, the physician will attach the ChARM device, and the research staff will start videorecording. Once the child is calm, the physician will start the ChARM device and also begin counting RR manually, and the videorecording will be done in parallel to the physician's use of ChARM like before. The physician will count the total number of breaths until ChARM completes its counting. The ChARM device will show a green or red signal when it finishes its count. The videorecording will continue until the device is done. The time taken by the ChARM device to get the RR will be recorded. A maximum of 3 min will be allowed for the ChARM to register a reading or display an error message. Otherwise, it will be documented as a failed attempt. The physician will make a maximum of three attempts for each manual count and ChARM count.

Each video file will be saved in the camera with a file name. The research staff will note the file name in the data collection sheet. Then the videos will be downloaded from the camera and saved on a password-protected computer and the videos will be deleted the camera. After that he will edit the video by removing additional portions of the recording, removing sound or identifiable features, and reducing file size. Video Editing Software (Adobe Premiere Pro) will be used for video editing.

There will be two expert paediatricians (ie, medical practitioners having a postgraduate degree in paediatrics and specialised in treating sick children). These videos will be sent to them for their feedback on video quality (decision of using tripod, camera positions, angles, light or other issues) and to count RR. Their feedback will be incorporated, and the videorecording procedures will be modified accordingly. An SOP for videography will be developed to maintain consistency across all study hospitals/clinics.

Figure 1 shows the schematic diagram of developing the SOP for videography.

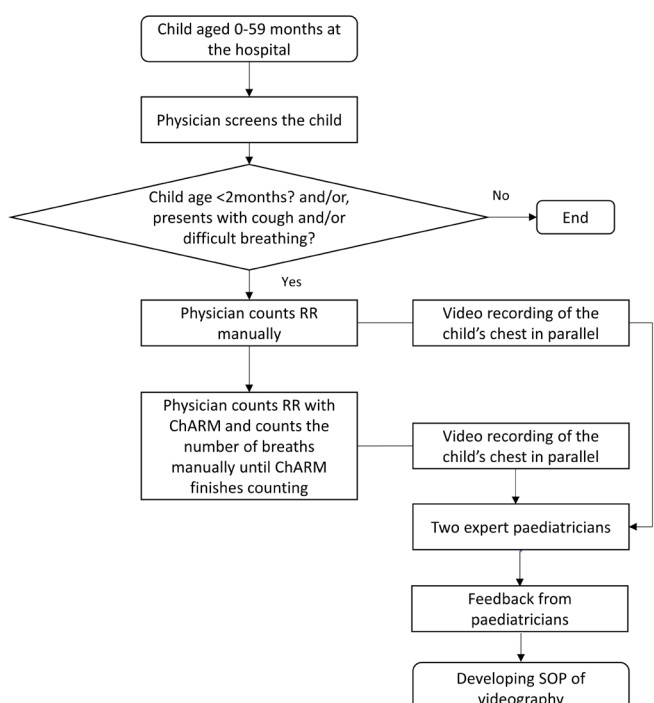

**Figure 1** Developing SOP of videography. ChARM, Children's Automated Respiration Monitor; SOP, standard operating procedure .

## Making a set of reference videos

Following the development of the SOP, videos will be recorded from 50 children using this standardised procedure. A physician will count the RR manually once and with the ChARM device once, and videorecording will be done simultaneously. The same procedure described in figure 1 will be followed. If the physician fails in their first attempt to have a satisfactory recording, they will make two more attempts for each manual and ChARM counts. The reason for the attempted failure will be documented. The condition of the child during RR measurement will also be noted.

There will be three expert paediatricians. Each video will randomly be allocated to two of the paediatricians. If both paediatricians disagree, that is, disagreement in readability or difference of RR>±2 breaths per minute (bpm), then the video will be sent to the third paediatrician. If any two paediatricians agree, that is, agreement in interpretability and difference of RR≤±2 bpm, the videos will be considered reference videos. The mean RR count of paediatricians in agreement will be considered as the final RR count.

Figure 2 presents the schematic diagram of making a set of reference videos.

## Training and standardisation of VEP members

Six local physicians with a Bachelor of Medicine and Bachelor of Surgery(MBBS) degree will be invited for training and standardisation. The training session will be conducted using an online platform. The physicians will be trained on counting RR using videos of known RR

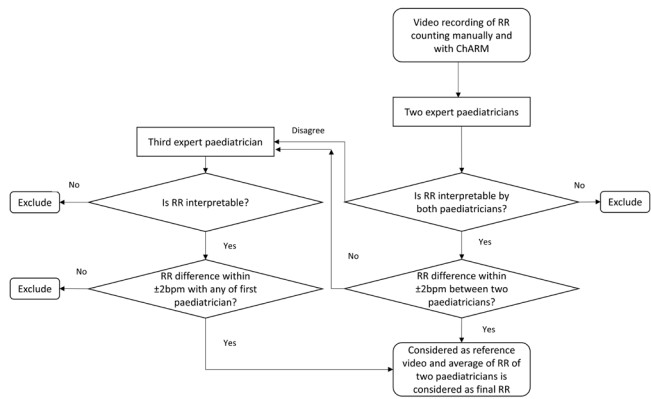

**Figure 2** Making a set of reference videos. ChARM, Children's Automated Respiration Monitor, RR, respiratory rate.

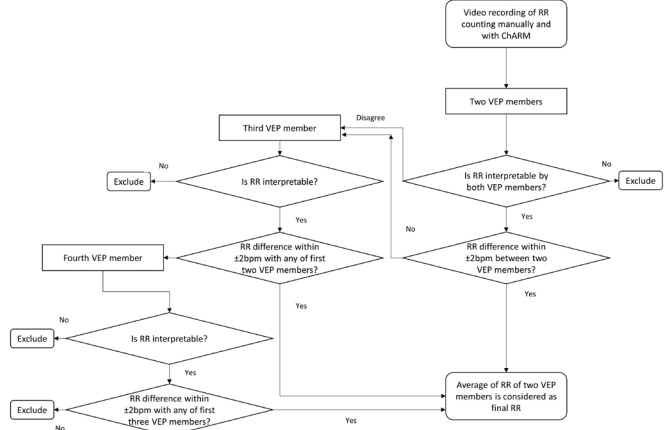

**Figure 3** Respiratory rate interpretation from videorecording by video expert panel. ChARM, Children's Automated Respiration Monitor, RR, respiratory rate; VEP, video expert panel.

counts with different level of breathing from reference videos. Training will be provided following the WHO Integrated Management of Childhood Illness guidelines.[4 5] The principal investigator (PI) will conduct the training sessions. After completion of training, they will be evaluated individually in a blinded manner. Twenty videos with different level of breathing will be assigned to each VEP member from the reference videos. The RR interpreted by each VEP member will be considered accurate if this is within±2 bpm of the RR interpreted by expert paediatricians. Those physicians who achieve a pass mark (at least 80%) in the evaluation will be considered to work as VEP members in the study.

### Evaluating the performance of VEP

The performance of VEP will be evaluated using the videos recorded in different levels of health facilities (CCs, subdistrict hospital and ICMH). RR of each child will be assessed twice, that is, manually and using ChARM by the CHCP in the CC and by a physician in the hospital (as described in figure 1). The videos will be recorded following the SOP. Each video will be randomly assigned to two VEP members to interpret the RR. If there is an agreement of being uninterpretable, the videos will be excluded. If the RR is≤±2 bpm between both members, their average RR will be considered final. However, if there is a disagreement in interpretability or the difference of the RR count is >2 bpm between the first two members, the video will be sent to a third-panel member. If there are inconsistencies among all three members (if the RR is not ≤2 bpm between any two members), the videos will be sent to a fourth-panel member. If there is agreement between the fourth-panel member and any other member who read the video earlier, their average RR will be considered final. However, if there is no agreement between any two panel members, the videos will be excluded from the analysis (figure 3).

### Quality control measures

The videos will be checked periodically if they adhere to the SOP to ensure quality by the PI. Any quality issues will

be reported, and corrective feedback will be provided. The expert paediatricians will view a 10% random sample of videos in which they are blinded to the panel's final classification. The panel's overall performance will be evaluated using the expert's interpretation as the reference. Lastly, to monitor individual member interpretation reproducibility, all members will randomly be reassigned 20% of images they interpreted previously (at least 1 week earlier) to check intrareader agreement.

### Development of a web-based automated system

A web-based automated system will be developed. This system will use the Hypertext Preprocessor platform for the user interface and the Structured Query Language (SQL) platform for the database. The purpose of developing this system is an automated distribution of the videorecordings among the expert paediatricians and VEP members, the input of readings by the members and the generation of automated reports. The system will be password protected and connected to a secured server. The expert paediatricians and VEP members will use this programme. Each will need to log in with their user ID and password. Access to each expert paediatrician and VEP member will be limited to what they are assigned. When logged in, they will see the list of recordings assigned to them. They will open the video on the computers and interpret the RR. The VEP members will be blinded to other members' readings.

### Operational definitions
#### Interpretability

The video will be considered 'interpretable' when the expert paediatrician or VEP member will be able to view the child's chest movements and measure RR for the whole duration. On the other hand, a video will be counted as 'uninterpretable' if the physician will not be able to view and count RR for the whole duration of the recording. The videos can be unreadable for various reasons, for example, wrong position and angle of the

camera, inappropriate focus, poor lighting, inadequate exposure or if the child is agitated, moving, crying, etc.

## Agreement and disagreement

The agreement will be defined when two expert paediatricians or VEP members interpret the video as
- uninterpretable.
- Interpretable and the difference between RR count≤±2 bpm.

Disagreement is when
- one expert paediatrician/VEP member interprets the video as interpretable, and another interprets it as unreadable or
- the difference between their RR count is>2 bpm.

## Data collection and storage

Field data (age, sex, RR, child condition during assessment) will be collected in paper forms. After entry, data will be transferred to a password-protected server (SQL Server 2008 R2) located in Dhaka, Bangladesh, in real-time using internet connectivity. Recorded videos will be transferred to encrypted OneDrive/Dropbox cloud storage. The VEP members will interpret RR from recorded videos using the web-based automated software, and the data will also be stored on the server in real time. Deidentified data will be transferred and stored on the DataShare server at the University of Edinburgh, UK.

## Sample size calculation

The videos from the first 30 children will be used to establish the best procedure for capturing videos. This number was purposively chosen as a qualitative assessment of the videos will be done. We assume that 30 children would be enough to develop the SOP. However, we will consider recruiting more if after 30 children a consistent SOP has not been finalised. After finalisation of the SOP, the videos from 50 children will be used for making a set of reference videos to train and standardise VEP. We did not use any statistical formula to calculate this sample size. These children will be recruited from the IPD of a hospital. Based on our experience in this setting, we assume that we will get a significant number of children with fast breathing. However, we will consider recruiting further children if our target has not been achieved. To evaluate the performance of VEP, Bland-Altman's statistical methods for assessing agreement between two methods of clinical measurement are used.[29 30] A total of 226 interpretable videos of each method (manual and ChARM) are required, which will provide 90% power to assess agreement in RR counts between two VEP members, considering type-I error α=0.05, expected mean difference±0.5, expected SD 1.5 and maximum allowed difference 4. Assuming 80% interpretability of the videos and 20% failure to record videos, the required number of children is about 350.

## Statistical analysis

To standardise the VEP members, we will estimate the mean difference of RR counts and the percent agreement within±2 bpm between each VEP member and expert paediatricians. We will consider expert paediatricians' counts as the gold standard. To assess the performance of VEP members, we will produce Bland-Altman plots[29] to assess the agreement in RR counts and calculate the percent agreement between primary panel members. We will calculate mean difference of RR counts and the per cent agreement within±2 bpm to assess the intrareader agreement of each VEP member. We will use Cohen's kappa statistic to estimate the inter-reader agreement of identifying fast breathing between panel members. We will also measure the mean difference of RR counts and the percent agreement within±2 bpm between each VEP member and expert paediatricians from QC data. To evaluate the accuracy of the ChARM device and manual counts, we will produce Bland-Altman plots to assess the agreement in RR counts of ChARM counts, and manual counts with VEP counts. We will measure the mean difference of RR counts and the percent agreement within±2 bpm. We will also use Cohen's kappa statistic to assess the agreement in identifying fast breathing between these methods. The mean time to measure the RR by the ChARM device will be assessed. To assess the potential influence of the ChARM device on the RR of the children, we will calculate the mean difference of RR between the first and second measurements interpreted by the VEP.

## ETHICS AND DISSEMINATION

Ethical approval was obtained from the National Research Ethics Committee of Bangladesh Medical Research Council, Bangladesh (Registration Number: 39315022021), and Edinburgh Medical School Research Ethics Committee (EMREC), Edinburgh, UK (REC Reference: 21-EMREC-040). Informed written consent will be taken from the parent or guardian of each child. Dissemination will be through conference presentations and publications in peer-reviewed journals. Anonymised data files will also be stored securely in the DataStore repository at the University of Edinburgh, UK and will be shared after the publication of the main paper.

**Author affiliations**
[1]Usher Institute, The University of Edinburgh, Edinburgh, UK
[2]Projahnmo Research Foundation, Dhaka, Bangladesh
[3]Department of Paediatrics, Johns Hopkins School of Medicine, Baltimore, Maryland, USA
[4]Department of International Health, Johns Hopkins University Bloomberg School of Public Health, Baltimore, Maryland, USA
[5]Department of Global Public Health, Karolinska Institute, Stockholm, Sweden
[6]Department of Paediatrics, Shaheed Suhrawardi Medical College Hospital, Dhaka, Bangladesh
[7]Royal Hospital for Children, Glasgow, UK
[8]Department of Neonatology, Institute of Child and Mother Health, Dhaka, Bangladesh
[9]Department of Paediatric Respiratory Medicine, The University of Edinburgh Centre for Inflammation Research, Edinburgh, UK

**Acknowledgements**  The authors are grateful for the support and contributions of Dr Arunangshu Dutta Roy, Asim Nehal, Rizouan Ur Rashid, Rakib Bhuiyan, Dr Nusrat Sharmin Asma, Dr Farjana Hossain, Dr Muhammad Shariful Islam, Dr Md. Jahurul Islam, and the Ministry of Health and Family Welfare, Government of Bangladesh. The authors also thank RESPIRE collaboration for their contribution in bringing the manuscript to its final shape. The RESPIRE collaboration comprises the UK Grant holders, Partners and research teams as listed on the RESPIRE website (www.ed. ac.uk/usher/respire) including Linda Bauld.

**Contributors**  AMK and HC conceptualised and designed this study. HC, SC, AB, EDM and TS provided mentorship to AMK. HC, SC, EDM and CK critically reviewed the study design. AMK, SA, MMR will be involved in project management. KN, RS and AA will review the videos and provide feedback to video quality. NHC and MSI will be responsible for data management. AMK drafted the manuscript, and all authors critically reviewed and approved the final manuscript before submission.

**Funding**  This research was funded by the UK National Institute for Health Research (NIHR) (Global Health Research Unit on Respiratory Health (RESPIRE); 16/136/109) using UK aid from the UK Government to support global health research.

**Disclaimer**  The views expressed in this publication are those of the author(s) and not necessarily those of the NIHR or the UK Government.

**Competing interests**  None declared.

**Patient and public involvement**  Patients and/or the public were not involved in the design, or conduct, or reporting, or dissemination plans of this research.

**Patient consent for publication**  Not applicable.

**Provenance and peer review**  Not commissioned; externally peer reviewed.

**ORCID iDs**
Ahad Mahmud Khan http://orcid.org/0000-0002-4347-0825
Salahuddin Ahmed http://orcid.org/0000-0001-6771-0638
Eric D McCollum http://orcid.org/0000-0002-1872-5566
Ting Shi http://orcid.org/0000-0002-4101-4535

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
