## [Reviewer comments · BMJ Open]

ARTICLE DETAILS

TITLE (PROVISIONAL)	Developing a video expert panel as a reference standard to evaluate respiratory rate counting in paediatric pneumonia diagnosis: protocol for a cross-sectional study
AUTHORS	Khan, Ahad; Ahmed, Salahuddin; Chowdhury, Nabidul Haque; Islam, Md Shafiqul; McCollum, Eric D.; King, Carina; Shi, Ting; Nahar, Kamrun; Simpson, Robynne; Ahmed, Ayaz; Rahman, Md Mozibur; Baqui, Abdullah; Cunningham, Steven; Campbell, Harry

VERSION 1 – REVIEW

REVIEWER	Hartley, Caroline University of Oxford, Department of Paediatrics
REVIEW RETURNED	24-Aug-2022

GENERAL COMMENTS	The authors describe a protocol for developing a video expert panel (VEP) as a standard for respiratory rate counting. The paper is generally well written with a clear description of the protocol. There are a few points though where I think further clarification is needed: 1) It is unclear how the VEP would be used as a reference standard once this study has been performed. While the method would be rigorously tested following this study, how would this group of people/videos be used to evaluate a new method of measuring RR if one were to be developed? Please could you add a description? Ideally an automated approach would be used in a hospital as this will resolve the issues discussed in the introduction relating to subjectivity and difficulties in measuring RR and may also be quicker. How will a new automated approach be assessed compared with the videos as a reference standard?2) I could not see a description of how the secondary objectives will be assessed/statistically analysed. Please include this.3) Exclusion criteria ii should state that the parents are unwilling to provide consent.4) In the section 'developing SOP of videography' it says that videos will be recorded from different camera positions - will several cameras be used in each test so that there are several camera angles, or will the angle be varied between children? Please clarify in the text. Also in this section it says that the video recording will be stopped when the ChARM device has finished counting - does this device count a certain number of breaths? Please add further details.5) Please add more clarification for how the sample size was chosen. For developing the SOP a sample of 30 is given, but what happens if after 30 infants a consistent SOP has not been developed? Or could it be possible to develop a standard SOP that
--

	is easy to use with fewer than 30 infants? How will you know when the SOP is finalised - please add clarification on this. 6) In developing the reference videos 50 children will be recorded. Please clarify why this number was chosen. Also, clarify how you will ensure that different severity of illness/levels of breathing (e.g. normal breathing, slow, fast) are included? What happens if all 50 children have normal breathing? 7) It isn't clear why the physician counts RR manually during recordings or why the ChARM device is used. Please can you add further details relating to this - is the VEP RR count compared to these other measures? Is the manual count compared with the ChARM output? Are these counts just used to ensure the videos are the correct length?
--	--

REVIEWER	Contesse, Marielle Casimir, Research
REVIEW RETURNED	26-Aug-2022

GENERAL COMMENTS	This study protocol describes research to develop a video expert panel as a reference standard to evaluate respiratory rate counting for identifying pneumonia in children. The authors clearly describe why this reference standard is necessary in the Introduction, and the research has been carefully designed to meet the study objectives. I have provided some suggestions for strengthening the study procedures section below: Study Procedures:  1. In the first paragraph of Developing SOP of videography, can the authors clarify if there are pre-set different camera positions, angles, levels of light, and tripod use that will be tested? Or are they saying that there will be variability in those aspects of the recording that they are not controlling? The paragraph was confusing as it is currently worded. 2. In the second paragraph, the authors state that a maximum of 3 minutes will be allowed for the ChARM to register a reading or display an error message. If there is a failed attempt, would they continue to try until they have a successful attempt? 3. Are the cameras that will be recording the videos compliant with national standards to protect sensitive patient health information? Will the videos be deleted from the cameras after the videos have been uploaded to the secure computers? 4. In the Training and Standardisation of VEP Members section, can the authors go into more detail on how the local physicians will be trained? In person or remotely? Can the authors describe how they are defining accuracy for the testing of each VEP member? Is it RR $\leq \pm 2$ bpm? 5. In the Quality Control Measures section, can the authors describe whether there are specific quality review criteria that the PI is evaluating? 6. In the Quality Control Measures section, can the authors include the amount of time between interpretation of repeat images when determining intra-reader agreement? Will they wait a certain number of days, weeks, months between each interpretation as a standard?
--

	7. Can the authors clarify whether the development of the web-based automated system will be used for the evaluation and performance of the VEP or if it will be developed later? 8. In the Data Collection and Storage section, can the authors describe what is included in “field data”? 9. In the Sample Size Calculation section, can the authors describe how 30 was selected for the first phase and 50 was selected for the second phase? 10. In the Statistical Analysis section, can the authors describe the statistical methods for calculating intra-reader agreement? Study Limitations 11. In the Strengths and Limitations section, can the authors elaborate on study limitations? The bullet points provide background information rather than strengths and limitations. It might help to break these up into two sections, since it is not clear which are strengths and which are limitations.
--	---

VERSION 1 – AUTHOR RESPONSE

Reviewer 1 Comments:

The authors describe a protocol for developing a video expert panel (VEP) as a standard for respiratory rate counting. The paper is generally well written with a clear description of the protocol. There are a few points though where I think further clarification is needed:

1) It is unclear how the VEP would be used as a reference standard once this study has been performed. While the method would be rigorously tested following this study, how would this group of people/videos be used to evaluate a new method of measuring RR if one were to be developed? Please could you add a description? Ideally an automated approach would be used in a hospital as this will resolve the issues discussed in the introduction relating to subjectivity and difficulties in measuring RR and may also be quicker. How will a new automated approach be assessed compared with the videos as a reference standard?

Response: In this study, we will establish a standard operating procedure (SOP) for capturing quality videos, make a set of reference videos, and train and standardise the VEP members using the reference videos. The SOP, recorded videos, and evaluation methods could be used for evaluating new RR counting devices. We have clarified this in the Introduction section (page 4).

2) I could not see a description of how the secondary objectives will be assessed/statistically analysed. Please include this.

Response: Thank you for pointing out this. To evaluate the accuracy of the ChARM device and manual counts, we will produce Bland-Altman plots to assess the agreement in RR counts of ChARM counts, and manual counts with VEP counts. We will measure the mean difference of

RR counts and the percent agreement within ± 2 bpm. We will also use Cohen's kappa statistic to assess the agreement in identifying fast breathing between these methods. The mean time to measure the RR by the ChARM device will be assessed. To assess the potential influence of the ChARM device on the RR of the children, we will calculate the mean difference of RR between the first and second measurements interpreted by the VEP. We have included this in the revised manuscript (page 9).

3) Exclusion criteria ii should state that the parents are unwilling to provide consent.

Response: This has been revised accordingly (page 5).

4) In the section 'developing SOP of videography' it says that videos will be recorded from different camera positions - will several cameras be used in each test so that there are several camera angles, or will the angle be varied between children? Please clarify in the text.

Response: The videography procedure (e.g., camera positions, angles and levels of light, use of tripod etc.) will be varied between children depending on the child and environmental factors (e.g., child condition, position, adequate exposure, place of assessment etc.). This has been clarified (page 6).

Also in this section it says that the video recording will be stopped when the ChARM device has finished counting - does this device count a certain number of breaths? Please add further details.

Response: ChARM automatically counts the number of respiration for a particular period and calculates the respiratory rate for one minute. This has been added in the introduction section (page 4).

5) Please add more clarification for how the sample size was chosen. For developing the SOP a sample of 30 is given, but what happens if after 30 infants a consistent SOP has not been developed? Or could it be possible to develop a standard SOP that is easy to use with fewer than 30 infants? How will you know when the SOP is finalised - please add clarification on this.

Response: The sample of 30 children was purposively chosen as the qualitative assessment of the videos will be done. We assume that 30 children would be enough to develop the SOP. However, we will consider more if after 30 children a consistent SOP has not been finalized. We have clarified this in the revised manuscript (page 9).

6) In developing the reference videos 50 children will be recorded. Please clarify why this number was chosen. Also, clarify how you will ensure that different severity of illness/levels of breathing (e.g. normal breathing, slow, fast) are included? What happens if all 50 children have normal breathing?

Response: We did not use any statistical formula to calculate this sample size of 50 children. These children will be recruited from the inpatient department (IPD) of a hospital. Based on our experience in this setting, we assume that we will get a significant number of children with fast breathing. However, we will consider recruiting further children if our target has not been achieved. We have clarified this in the revised version (page 9).

7) It isn't clear why the physician counts RR manually during recordings or why the ChARM device is used. Please can you add further details relating to this - is the VEP RR count compared to these other measures? Is the manual count compared with the ChARM output? Are these counts just used to ensure the videos are the correct length?

Response: We will assess the accuracy of the ChARM device in counting RR compared to the VEP RR. We will also assess the agreement between manual RR counts by healthcare staff and VEP RR counts. We have added this as a secondary objective in the revised version (page 5).

Reviewer 2 Comments:

This study protocol describes research to develop a video expert panel as a reference standard to evaluate respiratory rate counting for identifying pneumonia in children. The authors clearly describe why this reference standard is necessary in the Introduction, and the research has been carefully designed to meet the study objectives. I have provided some suggestions for strengthening the study procedures section below:

Study Procedures:

1. In the first paragraph of Developing SOP of videography, can the authors clarify if there are pre-set different camera positions, angles, levels of light, and tripod use that will be tested? Or are they saying that there will be variability in those aspects of the recording that they are not controlling? The paragraph was confusing as it is currently worded.

Response: The videography procedure (e.g., camera positions, angles and levels of light, use of tripod etc.) will be varied between children depending on the child and environmental factors (e.g., child condition, position, adequate exposure, place of assessment etc.). We have clarified this in our revised version (page 6).

2. In the second paragraph, the authors state that a maximum of 3 minutes will be allowed for the ChARM to register a reading or display an error message. If there is a failed attempt, would they continue to try until they have a successful attempt?

Response: The physician will make a maximum of three attempts for each manual and ChARM count. If failed, this will be considered a failure of RR measurement. We have clarified this in our revised version (page 6).

3. Are the cameras that will be recording the videos compliant with national standards to protect sensitive patient health information? Will the videos be deleted from the cameras after the videos have been uploaded to the secure computers?

Response: We will use Canon EOS M50 camera in our study. This camera does not collect any information sensitive patient health information. We will not record the child's face or any other

identifiable part. Videos will be deleted from the camera after transferring those to a secure computer. We have added this in our revised version (page 6).

4. In the Training and Standardisation of VEP Members section, can the authors go into more detail on how the local physicians will be trained? In person or remotely? Can the authors describe how they are defining accuracy for the testing of each VEP member? Is it $RR \leq \pm 2$ bpm?

Response: The training session will be conducted using an online platform. The RR interpreted by each VEP member will be considered accurate if this is within ± 2 bpm of the RR interpreted by expert paediatricians. We have provided more information on training in the revised manuscript (page 7).

5. In the Quality Control Measures section, can the authors describe whether there are specific quality review criteria that the PI is evaluating?

Response: The videos will be checked periodically if they adhere to the SOP to ensure quality by the PI. We have added this in the revised manuscript (page 8).

6. In the Quality Control Measures section, can the authors include the amount of time between interpretation of repeat images when determining intra-reader agreement? Will they wait a certain number of days, weeks, months between each interpretation as a standard?

Response: They will get the repeat image after at least one week. We have added this in this section (page 8).

7. Can the authors clarify whether the development of the web-based automated system will be used for the evaluation and performance of the VEP or if it will be developed later?

Response: We have mentioned that the expert paediatricians and VEP members will use this program in this study (page 8).

8. In the Data Collection and Storage section, can the authors describe what is included in "field data"?

Response: Field data include age, sex, RR, child condition during assessment. These have been added in the revised version (page 9).

9. In the Sample Size Calculation section, can the authors describe how 30 was selected for the first phase and 50 was selected for the second phase?

Response: For the first phase, the sample of 30 children was purposely chosen as the qualitative assessment of the videos will be done. We assume that 30 children would be enough to develop the SOP. However, we will consider more if after 30 children a consistent SOP has not been finalized. For the second phase, We did not use any statistical formula to calculate this sample size of 50 children. These children will be recruited from the inpatient department (IPD) of a hospital. Based on our experience in this setting, we assume that we will get a significant

number of children with fast breathing. However, we will consider recruiting further children if our target has not been achieved. We have clarified this in the revised version (page 9).

10. In the Statistical Analysis section, can the authors describe the statistical methods for calculating intra-reader agreement?

Response: We will calculate the mean difference of RR counts and the percent agreement within ± 2 bpm to assess the intra-reader agreement of each VEP member. This has been added to the revised manuscript (page 9).

Study Limitations

11. In the Strengths and Limitations section, can the authors elaborate on study limitations? The bullet points provide background information rather than strengths and limitations. It might help to break these up into two sections, since it is not clear which are strengths and which are limitations.

Response: According to the guideline of BMJ Open, it is not possible to break these up into two sections. Therefore, we have specified the limitation in the revised version (page 3).

We have made a minor correction in Figure 3 and uploaded the updated one.

Thank you very much for your consideration.

VERSION 2 – REVIEW

REVIEWER	Hartley, Caroline University of Oxford, Department of Paediatrics
REVIEW RETURNED	24-Oct-2022

GENERAL COMMENTS	Thank you for addressing my comments, I wish you all the best with this study.
--

REVIEWER	Contesse, Marielle Casimir, Research
REVIEW RETURNED	11-Oct-2022

GENERAL COMMENTS	All of my comments have been addressed.
---